# Enhancements in yield, agronomic, and zinc recovery efficiencies of rice-wheat system through bioactive zinc coated urea application in Aridisols

**Syed Shahid Hussain Shah[1], Muhammad Azhar** [1]*, **Faisal Nadeem[2], Muhammad Asif Ali[1], Muhammad Naeem Khan[3], Ijaz Ahmad[1], Muhammad Yasir Khurshid[1], Muhammad Hasnain[1], Zeeshan Ali[1], Ahmad Abu Al-Ala Shaheen[1]**

**1** Department of Agronomy, Engro Fertilizers Ltd., Lahore, Pakistan, **2** Department of Soil Science, University of the Punjab, Lahore, Pakistan, **3** Directorate General Soil Survey of Punjab, Agriculture Department, Lahore, Pakistan

* azharuaf509@gmail.com, mazhar@engro.com

**Data Availability Statement:** All relevant data and information are present in the article.

## Abstract

### Background

Zinc (Zn) deficiency and source-dependent Zn fertilization to achieve optimum Zn levels in rice and wheat grains remain global concern for human nutrition, especially in developing countries. To-date, little is known about the effectiveness of bioactive Zn-coated urea (BAZU) to enhance the concentration, uptake, and recovery of Zn in relation to agronomic efficiency in paddy and wheat grains.

### Results

Field experiments were carried out during 2020–21 on the rice-wheat system at Lahore, Faisalabad, Sahiwal, and Multan, Punjab, Pakistan using four treatments viz.$T_1$ (Urea 46% N @ 185 kg ha$^{-1}$ + zero Zn), $T_2$ (Urea 46% N @ 185 kg ha$^{-1}$ + ZnSO$_4$ 33% Zn @ 15 kg ha$^{-1}$), $T_3$ (BAZU 42% N @ 103 kg ha$^{-1}$ + Urea 46% N @ 62 kg ha$^{-1}$ + 1% bioactive Zn @ 1.03 kg ha$^{-1}$) and $T_4$ (BAZU 42% N @ 125 kg ha$^{-1}$ + Urea 46% N @ 62 kg ha$^{-1}$ + 1% bioactive Zn @ 1.25 kg ha$^{-1}$) in quadruplicate under Randomized Complete Block Design. Paddy yield was increased by 13, 11, 12, and 11% whereas wheat grain yield was enhanced by 12, 11, 11, and 10% under $T_4$ at Multan, Faisalabad, Sahiwal, and Lahore, respectively, compared to $T_1$. Similarly, paddy Zn concentration was increased by 58, 67, 65 and 77% (32.4, 30.7, 31.1, and 34.1 mg kg$^{-1}$) in rice whereas grain Zn concentration was increased by 90, 87, 96 and 97% (46.2, 43.9, 46.7 and 44.9 mg kg$^{-1}$) in wheat by the application of BAZU ($T_4$) at Multan, Faisalabad, Sahiwal, and Lahore, respectively, in comparison to $T_1$. Zinc recovery was about 9-fold and 11-fold higher in paddy and wheat grains, respectively, under BAZU ($T_4$) treatment relative to $T_2$ while, the agronomic efficiency was enhanced up to 130% and 141% in rice and wheat respectively as compared to $T_2$.

**Funding:** Experimental funds were provided by Engro Fertilizers Limited. There is no existence of any conflict of interest in this study. The research work is conducted by the research and development (R&D) section, agronomy department of Engro Fertilizers Limited. Adhering strongly with the research ethics, this wing already has research publications in renowned scientific journals. In this study, we did not advocate the superiority of our product, rather, we used the term BAZU instead of our product's trade name which removes any conflict of interest. Moreover, the abbreviation of BAZU stands for bioactive Zn-coated urea which is not a name of any of Engro's products.

**Competing interests:** This manuscript is original research and not submitted or published previously and it is not under consideration for publication elsewhere. The publication has been approved by all co-authors with no competing/conflicting interests.

## Conclusion

Thus, $T_4$ application at the rate of 125 kg ha$^{-1}$ could prove effective in enhancing the rice paddy and wheat grain yield along with their Zn biofortification ($\sim$34 mg kg$^{-1}$ and $\sim$47 mg kg$^{-1}$, respectively) through increased agronomic and Zn recovery efficiencies, the underlying physiological and molecular mechanisms of which can be further explored in future.

## Introduction

Zinc (Zn) is a vital micronutrient and its deficiency in food crops and malnutrition problem in humans are continuously increasing in developing countries including Pakistan [1, 2]. The Zn deficiency in humans is mainly responsible for immunity dysfunction, pregnancy complications, impair healthy growth of babies, and vulnerability to other diseases [3, 4]. Up to 22% of Pakistanis are prone to Zn malnutrition [5, 6]. Children, pregnant and breastfeeding women need higher Zn and are more prone to its malnutrition. The daily Zn requirement of an adult ranges from 8–11 mg while pregnant/lactating women need 11–13 mg of Zn per day [7]. About 10% (estimated as three thousand proteins) of human body proteins are Zn-dependent [8, 9]. Cereal crops such as wheat, rice, and maize are major staple foods worldwide, and more than half of the global population is dependent on wheat [10] and rice [11] for their daily dietary intake. More than 60% of the total Zn requirement, for the human body, has been achieved through cereal staple foods in South Asian countries including Pakistan [12, 13]. Higher consumption of these cereal crops having lower bioavailable Zn content is a major reason for malnutrition [2, 6]. Currently, the cultivars of cereals are unable to meet nutritional requirements due to their lesser inherited Zn concentration and more than 50% of global soils used for wheat cultivation are deficient in phyto-available Zn [2]. Zn deficiency prevails in about three billion people in the world and results in loss of about half million lives annually [14]. More than 4% of the worldwide mortality and morbidity in children under five and 16 million of the global disability-adjusted life years are caused by Zn deficiency [15, 16]. Deficiencies of Zn and other micronutrients in developing countries are also reported to cause great economic losses and have a considerable effect on the gross national product by decreasing productivity and increasing the health care costs [17, 18].

Although a majority of crop plants are vulnerable to Zn deficiency, rice is more sensitive as compared to other crops [19–21] because Zn is directly or indirectly responsible to activate enzymes, protein formation, metabolism of nucleic acid, and starch involved in pollination [2, 22, 23]. Zn is involved in photosynthesis, sugar transformation [24], flowering, and grain formation [25]. Zn deficiency affects fertilization in plants by altering stigma and pollen grains functioning and by affecting pollen viability [26]. Application of $ZnSO_4$ and Zn enriched urea increases grain yield in wheat [27–29] and rice [28, 30]. Paddy yield, Zn recovery, and agronomic efficiencies are improved by the application of Zn [31]. The availability of Zn is affected by several soil factors including pH, redox potential, and soil solution concentration of Zn, P, Mn, and Fe [32–34]. For example, Zn precipitates as zinc sulfide (ZnS) in flooded, zinc hydroxide [35] in basic, and as zinc carbonate ($ZnCO_3$) in calcium carbonate-dominated soils [36], which minimizes phyto-availability of Zn [21, 37]. Magnesium to calcium ratio, bicarbonate, and organic matter are other soil properties affecting Zn phyto-availability [38–40].

Several options are under experimentation to attain the required Zn levels in grains of staple food crops. Among these, agronomic fortification of cereal grains is a cost-effective and viable option to enhance grain Zn levels and to minimize Zn-oriented nutritional

complications, especially in Asian countries including Pakistan that are dependent on staple foods [6, 31]. Grain fortification can be done through two approaches i.e. breeding [41–43] and Zn-fertilization [31]. The second approach is economical and easily applicable to improve grain Zn contents [41, 42]. There are few studies on the benefits of Zn-biofortification [31]. Previous study has reported the nutrient delivery in wheat through the application of dual-capped Zn-urea nano-fertilizers [44]. However, no study is reported to compare rice paddy/wheat grain Zn concentration and recovery between bioactive zinc coated urea, produced through Bioactive Nutrient Fortified Fertilizer (BNFF)[©] patent process [45], and $ZnSO_4$.

The BAZU is a synergetic hybrid of urea, Bioactive Zinc (BAZ)[©] and Bioactive Coating (BAC)[©]; a consortium of Zn and other nutrients solubilizing and mobilizing bacteria. BAZ[©] is organically encapsulates Zn that is less prone to fixation, sandwiching, and trapping in soil structure. BAZ[©] is gradually released in the rhizosphere as per plant demand that supports an uninterrupted and continuous supply of Zn during the crop cycle. In addition, BAC[©] enhances root growth, mobilizes other nutrients present in the rhizosphere, and triggers induced systemic resistance of plants to withstand stress conditions. Coating covers of BAZ[©] and BAC[©] encapsulate urea prills, induce a slow N release mechanism, contribute to reducing N losses and enhance N use efficiency. Collectively, BAZU is revolutionary fertilizer suitable for all types of soils, climates, and crops [45].

Bioactive Zn-coated urea is an emerging novel approach for the grain Zn fortification not tested to compare rice paddy/wheat grain for Zn concentration, Zn recovery, and agronomic efficiencies. Therefore, this study hypothesized whether BAZU can enhance paddy/grain yield, Zn concentration, and recovery in comparison to other Zn sources, primarily, due to long-term enhancements of Zn Phyto-availability. The objective of the present study was to evaluate the most efficient and cost-effective Zn source available to enhance rice-wheat yield and the paddy/grain Zn concentration.

## Materials and methods

### Site selection, soil analysis, and climatic conditions

The present study was carried out at four sites i.e. farmer's field in Multan (29˚.959593 N, 71˚.343759 E), Faisalabad (31˚.7053030 N, 73˚.0215580 E), Sahiwal (30˚.533018 N, 72˚.758652 E) and Lahore (31˚.748680 N, 74˚.103364 E) regions. Two-year experiments were conducted on rice (2019 and 2020) and wheat (2019–20 and 2020–21) separately at each site. Pre-sowing soil samples (0–15, 15–30 cm) were analyzed for pH, electrical conductivity (EC), phosphorus (P), potassium (K), zinc (Zn), boron (B), and texture. The pH, EC, B, and texture were measured by following the methods described by [46, 47]. Soil organic matter was determined following Walkley and Black method [48]. The soil of each site was classified according to the manual of the Soil Science Division Staff [49]. Soil-saturated paste was prepared for pHs, extract of paste was taken for ECe and both were determined using Jenway EC and pH meter model 671P. The P and K were determined using methods of [50, 51] respectively. The concentrations of AB-DTPA extractable Zn were determined following [52] method. Briefly, an extractant solution (AB-DTPA) was prepared by dissolving specified quantities of $NH_4HCO_3$ and DTPA in 1.0 L of distilled water. Soil (10 g) was taken in a calibrated plastic centrifuge tube, and a newly prepared extractant solution (20 mL) was added. The suspension was then shaken for 2 h and the solution was filtered and analyzed for Zn contents using an atomic absorption spectrophotometer (Solar S-100, Thermo Electron, USA). These soil properties are presented in Table 1.

Multan is the southern part of Punjab, and the climate is arid subtropical with extreme summer temperature. Mean winter and summer temperature ranges from 7–26˚C and 29–

**Table 1. Physio chemical properties of pre-sowing soil on different locations.**

| Region | Depth (cm) | pH | ECe (dS m$^{-1}$) | P (mg kg$^{-1}$) | K (mg kg$^{-1}$) | B (mg kg$^{-1}$) | Zn (mg kg$^{-1}$) | Texture | OM (%) | Soil groups |
|---|---|---|---|---|---|---|---|---|---|---|
| | | | | | Rice | | | | | |
| MTN | *0–15 (15–30) | 8.5 (8.6) | 3.66 (3.05) | 07 (02) | 153 (125) | 0.21 (0.19) | 1.04 (1.01) | Loam (Loam) | 0.42 (0.34) | Haplocambid |
| FSD | 0–15 (15–30) | 7.9 (7.9) | 2.31 (2.42) | 12 (12) | 62 (62) | 0.44 (0.41) | 0.32 (0.29) | Loam (Loam) | 0.51 (0.37) | Haplocambid |
| LHR | 0–15 (15–30) | 8.1 (8.1) | 2.15 (1.48) | 08 (07) | 162 (141) | 0.14 (0.17) | 0.41 (0.64) | Loam (Loam) | 0.56 (0.32) | Calciargid |
| SWL | 0–15 (15–30) | 8.5 (8.4) | 1.31 (1.14) | 03 (04) | 187 (125) | 0.50 (0.48) | 1.45 (1.48) | Loam (Loam) | 0.47 (0.34) | Calciargid |
| | | | | | Wheat | | | | | |
| MTN | 0–15 (15–30) | 8.2 (8.5) | 1.62 (1.31) | 3.0 (03) | 195 (162) | 0.39 (0.24) | 1.44 (1.38) | Loam (Loam) | 0.51 (0.36) | Haplocambid |
| FSD | 0–15 (15–30) | 8.4 (8.4) | 2.46 (2.20) | 6.0 (7.0) | 67 (61) | 0.58 (0.53) | 0.54 (0.47) | Loam (Loam) | 0.56 (0.33) | Haplocambid |
| LHR | 0–15 (15–30) | 8.5 (8.6) | 3.85 (3.19) | 5.0 (5.0) | 170 (165) | 0.19 (0.14) | 0.38 (0.33) | Loam (Loam) | 0.63 (0.42) | Calciargid |
| SWL | 0–15 (15–30) | 8.2 (8.0) | 2.90 (3.18) | 05 (04) | 180 (175) | 0.57 (0.49) | 1.71 (1.29) | Loam (Loam) | 0.54 (0.39) | Calciargid |

MTN = Multan; FSD = Faisalabad; SWL = Sahiwal; LHR = Lahore

* = Soil analysis results of 0–15 cm depth; () = values in bracket are soil analysis results of 15–30 cm depth; ECe = Electrical conductivity of soil saturated paste extract;

P = Phosphorus; K = Potassium; B = Boron; Zn = Zinc; OM = Organic matter.

51˚C respectively. The climate of Faisalabad is semiarid subtropical with a mean temperature of 6–21˚C in winter and 27–39˚C in summer. Sahiwal has a semiarid subtropical climate with average winter and summer temperature of 7–25˚C and 28–49˚C respectively. The climate of Lahore is semiarid subtropical, and temperature may range from 4–21˚C in winter and 25–39˚C in summer.

## Experimental design and treatment application

Rice seed variety super basmati was taken from Rice Research Institute (RRI) Kala Shah Kaku and wheat variety Faisalabad 2008 was obtained from Wheat Research Institute, Ayub Agricultural Research Institute Faisalabad. Experiments at each location for both years were laid out in randomized complete block design arrangements and treatment plots (20 m × 10 m) were replicated four times. Experiments were carried out at the same locations for (rice-wheat-rice-wheat) in both successive years but treatments were applied separately to each cropping season. Thirty days old nursery of rice was shifted in puddled flooded field plots. The field remained flooded (∼10 cm depth) for one week after seedling transplantation, drained after one week, and refilled (∼10 cm depth). Treatments were $T_1$ (Urea 46% N @ 185 kg ha$^{-1}$ + zero Zn), $T_2$ (Urea 46% N @ 185 kg ha$^{-1}$ + ZnSO$_4$ 33% Zn @ 15 kg ha$^{-1}$), $T_3$ (BAZU 42% N @ 100 kg ha$^{-1}$ + Urea 46% N @ 62 kg ha$^{-1}$ + 1% bioactive Zn @ 1.00 kg ha$^{-1}$), $T_4$ (BAZU 42% N @ 125 kg ha$^{-1}$ + Urea 46% N @ 62 kg ha$^{-1}$ + 1% bioactive Zn @ 1.25 kg ha$^{-1}$). Fertilizer application rates were used following the recommendations of the Directorate of Agricultural information Government of Punjab Pakistan, however, there are no recommendations for BAZU therefore BAZU was tested at two different levels. The BAZU is a synergetic combination of urea coated with bioactive Zn (BAZ) @ 1% and beneficial microbial consortium @ 10$^3$ CFU g$^{-1}$ (Patent number US 9,994,494 "Bioactive Nutrient Fortified Fertilizers (BNFF)" published by US Patent and Trademark Office and patent number 142829 published by The Patent Office, Government of Pakistan). Fertilizers were applied separately for both crops @ 86:74 P$_2$O$_5$:K$_2$O kg ha$^{-1}$ in the form of diammonium phosphate (DAP; 46% P$_2$O$_5$; 18% N) and muriate of potash (MOP; 60% K$_2$O). The full dose of P, K$_2$O was applied as basal while nitrogen in the form of urea (46% N) was applied in two equal splits at tillering and panicle initiation stage. Nitrogen in the form of bioactive Zn-coated urea (42% N + 1% bioactive Zn) for $T_3$ and $T_4$ was applied in a single split as the first urea and the remaining N was applied in the form of urea (46% N)

as second split. The full dose of $ZnSO_4$ in $T_2$ was applied with the first urea split. The field was irrigated after seven days to continue flooding until physical maturity. After rice harvesting, wheat (var. FSD-08) seed @ 125 kg ha$^{-1}$ was broadcasted in similar soil at field capacity moisture. The $ZnSO_4$ (Zn = 33%) granular was purchased from the local market of Lahore city and imported from Kirns chemical Ltd. China with CAS (chemical abstracts service) number 7446-19-7 and EINECS (European inventory of existing commercial chemical substances) number 231-793-3. The BAZU was procured from the local market of Lahore with the brand name "Zabardast urea" marketed by Engro Fertilizers Pvt. Ltd. Pakistan. The Zabardast urea is a synergetic hybrid of urea having bioactive Zn @ 1% in 50 kg bag, nitrogen @ 42%, Zn mobilizing bacterial count @ $10^3$ CFU per gram of fertilizer material having 10% Zn mobilizing efficiency. The PSQCA (Pakistan standard and quality control authority) number for Zabardast urea is 5336–2015.

## Growth and yield attributes

At harvesting, An area of 100 cm$^2$ was selected in duplicate (as technical replicates) from every biological replicate, 10 plants from each technical replicate were selected for the determination of panicle/spike length, number of grains per panicle/spike of rice/wheat and the data were averaged to serve as one biological replicate [31, 53]. For paddy/grain and straw yield, three samples (1 m$^2$) were manually harvested and threshed to measure grain and straw weight. For 1000 paddy/grain weight, three samples of 1000 paddy/grain from each plot were taken and weighed. Plant height, Panicle/spike length were measured using stainless steel scale whereas the 1000 grains weight was measured by using a digital weighing balance (AUW 120 D, Shimadzu Corporation, Japan). Plant samples of both rice and wheat crops were taken following standard protocols with the permission of host farmers because experiments were conducted in farmer fields. The Harvest index (HI) of both crops was calculated as:

$$HI(\%) = \frac{\text{Grain yield}}{\text{Biological (Grain + Straw) yield}}$$

## Zn determination in grains/paddy

One gram of dried ground sample of paddy/grains of rice/wheat were processed for wet digestion as described by [35]. The samples were mixed separately with a 10 mL mixture of concentrated $HNO_3$ and $HClO_4$ (3:1) in a conical flask and kept overnight. The digestion was done using hot plate until a clear material was obtained. After cooling, samples were diluted to 50 ml with deionized water, filtered using Whatman filter paper 42, and stored in plastic bottles at room temperature (25 ± 2˚C). Digested samples were analyzed for Zn by atomic absorption spectrophotometer (Solar S-100, Thermo electron, USA) pre-calibrated with a series of Zn standard solutions.

Total Zn uptake by grains was calculated as follows:

$$Znuptake(gha^{-1}) = \text{Zn concentration in grain } (mgkg^{-1}) \times \text{Grain yield } (tha^{-1})$$

The Zn efficiencies i.e. Agronomic efficiency and Apparent Zn recovery efficiency were calculated by following [54].

Agronomic efficiency (AGE) was calculated as:

$$AGE(kgkg-1) = \frac{\text{Grain yield of Zn fertilized plants} - \text{Grain yield of control plants}}{\text{Quantity of Zn applied}}$$

The apparent recovery efficiency (ARE) of Zn was calculated as:

$$ARE(\%) = \frac{\text{Zn uptake in grains of fertilized plants} - \text{Zn uptake in grains of control plants}}{\text{Quantity of Zn applied}}$$

## Economic analysis

The economic analysis was conducted to estimate the net benefit of applied treatments. For this, the total (fixed and variable) cost and gross income (grain + straw) of both crops was averaged for two years. The detail of cost and income is given in Tab 4. The benefit to cost ratio (BCR) was measured following [55] as:

$$BCR = \frac{B}{C}$$

## Statistical analysis

The recorded parameters of both rice and wheat were statistically analyzed following two-way ANOVA [56] with randomized complete block design arrangements. The present study includes four treatments, four replications, four experimental sites, and two test crops (rice-wheat). The ANOVA was applied to each crop separately. To compare mean values, Least Significant Difference (LSD) test was applied at 5% probability using Statistix 8.1 software (Version 8.1 Software package).

# Results

## Rice

**Growth and yield attributes.** Results of Panicle length (PL), number of grains per panicle (GP), 1000 grain weight, biomass yield, and harvest index (HI) are presented in Table 2. PL

**Table 2. Effect of zinc sulfate and bioactive zinc coated urea on growth and yield of rice.**

| Region | Treatment | Panicle Length (cm) | No. Of Grains per panicle | 1000 grain weight (g) | Biomass yield (t ha$^{-1}$) | Harvest Index (%) |
|---|---|---|---|---|---|---|
| Multan | T1 | *22.0±0.15[h] | 69±1.53[gh] | 21.87±0.26[h] | 09.58±0.16[i] | 41.5±0.28[bc] |
| | T2 | 22.4±0.23[h] | 76±2.65[fg] | 22.65±0.32[g] | 09.92±0.18[i] | 42.4±0.12[ab] |
| | T3 | 22.0±0.30[h] | 76±2.08[fg] | 22.58±0.17[g] | 09.86±0.06[i] | 42.4±0.62[ab] |
| | T4 | 23.0±0.21[h] | 83±1.86[e] | 23.65±0.19[f] | 10.43±0.03[h] | 43.3±0.52[a] |
| Faisalabad | T1 | 25.9±0.23[fg] | 67±2.08[h] | 25.50±0.40[d] | 12.60±0.16[def] | 39.6±0.21[g] |
| | T2 | 25.7±0.40[g] | 73±3.21[fgh] | 26.23±0.18[b] | 13.05±0.33[bcd] | 40.3±0.12[defg] |
| | T3 | 25.5±0.23[g] | 74±2.65[fgh] | 26.07±0.18[bc] | 12.98±0.26[cd] | 40.9±0.06[cde] |
| | T4 | 26.6±0.34[efg] | 80±2.08[ef] | 27.33±0.15[a] | 13.46±0.24[ab] | 41.1±0.15[cd] |
| Sahiwal | T1 | 27.1±0.55[def] | 99±2.08[d] | 24.49±0.17[e] | 12.79±0.15[cd] | 39.8±0.05[fg] |
| | T2 | 27.7±0.49[de] | 107±2.52[c] | 25.50±0.20[d] | 13.21±0.06[bc] | 41.0±0.34[cd] |
| | T3 | 28.0±0.23[cd] | 106±1.73[cd] | 25.62±0.27[cd] | 13.13±0.06[bc] | 40.9±0.45[cde] |
| | T4 | 29.3±0.29[ab] | 113±2.33[bc] | 26.34±0.23[b] | 13.70±0.12[a] | 41.5±0.06[bc] |
| Lahore | T1 | 29.1±0.55[bc] | 107±2.08[c] | 23.43±0.25[f] | 11.89±0.25[g] | 39.9±0.23[efg] |
| | T2 | 29.4±0.43[ab] | 118±3.60[ab] | 24.37±0.20[e] | 12.29±0.02[efg] | 40.7±0.90[cdef] |
| | T3 | 29.8±1.01[ab] | 118±3.51[ab] | 24.36±0.18[e] | 12.18±0.24[fg] | 40.7±0.15[cdef] |
| | T4 | 30.6±1.15[a] | 122±2.64[a] | 25.36±0.17[d] | 12.75±0.23[cde] | 41.5±0.21[bc] |
| | LSD T×L | 1.246 | 7.001 | 0.492 | 0.461 | 1.000 |

*Average of two years data; ± Standard error; BAZU = Bioactive Zn coated urea (42% N; 1% Zn); T$_1$ (Urea 46% N @ 185 kg ha$^{-1}$+ zero Zn), T$_2$ (Urea 46% N @ 185 kg ha$^{-1}$ + ZnSO$_4$ 33% Zn @ 15 kg ha$^{-1}$), T$_3$ (BAZU 42% N @ 100 kg ha$^{-1}$ + Urea 46% N @ 62 kg ha$^{-1}$ + 1% bioactive Zn @ 1.00 kg ha$^{-1}$), T$_4$ (BAZU 42% N @ 125 kg ha$^{-1}$ + Urea 46% N @ 62 kg ha$^{-1}$ + 1% bioactive Zn @ 1.25 kg ha$^{-1}$); T = Treatment; L = Location.

was recorded higher under $T_4$ as compared to $T_1$ of the respective location at LHR (5% higher) and SWL (8% higher) while PL was similar among treatments at FSD and MTN. Application of BAZU ($T_4$) significantly improved GP at all locations as compared to $T_1$ (control) of respective location with a higher increment of 21% at MTN followed by FSD (19%), SWL (14%), and LHR (14%) over $T_1$ of each location. Among locations, higher GP were recorded at LHR followed by SWL, MTN, and FSD. Maximum biomass yield at MTN (9%), FSD (7%), SWL (7%), and LHR (7% higher) was attained under BAZU ($T_4$) relative to control ($T_1$) of respective locations whereas similar biomass yield was observed among $T_2$ and $T_3$ of each location. Similarly, higher paddy yield was achieved by Zn application through both sources at all experimental sites as compared to $T_1$ (control) of respective location, while increment in paddy yield was highest under $T_4$ (13, 11, 12 and 11% higher) over control (3.98, 4.98, 4.75 and 5.10 t $ha^{-1}$) of each location at MTN, FSD, LHR and SWL respectively (Fig 1). Although $T_2$ and $T_3$ showed higher paddy yield over control of respective locations but similar to each other. Among locations, the highest paddy yield was observed at SWL followed by FSD, LHR, and MTN (Fig 1). The treatments showed the following trend for paddy yield as $T_4 > T_3 = T_2 > T_1$ separately for each location. The weight of 1000 grain was significantly increased under $T_4$ by 7% at FSD and 8% at MTN, SWL, and LHR over $T_1$ (control) of respective locations but $T_2$ and $T_3$ showed a similar increase in 1000 grain weight relative to $T_1$ of the respective location. The $T_4$ showed a 4% higher harvest index (HI) at each experimental site compared to $T_1$ of the respective location. Except for SWL, similar HI was recorded among $T_2$ and $T_3$ relative to $T_1$ of each location (Table 2).

**Paddy Zn concentration and uptake.** Paddy Zn concentration and uptake were significantly increased with applied Zn at all experimental locations as compared to the control (Figs 2A and 3A). The following trend $T_4 > T_3 = T_2 > T_1$ was observed among treatments for paddy Zn concentration and uptake. Paddy Zn concentration under BAZU ($T_4$) fertilized plot was recorded 58% higher at MTN, 67% at FSD, 77% at LHR, and 65% at SWL experiments as compared to $T_1$ plants at MTN (20.5 mg $kg^{-1}$), FSD (18.4 mg $kg^{-1}$), SWL (18.8 mg $kg^{-1}$) and LHR (19.2 mg $kg^{-1}$) respectively (Fig 2A). Similarly, paddy Zn uptake was noted as highest with $T_4$ (146, 170, 180, and 177 g $ha^{-1}$) and lowest under control ($T_1$) plants (81, 92, 91and 96 g $ha^{-1}$) at MTN, FSD, LHR, and SWL respectively (Fig 3A).

**Apparent Zn recovery and agronomic efficiency.** Maximum paddy Zn was recovered under $T_4$ followed by $T_3$ and $T_2$ at all experimental sites (Fig 4A). The highest Zn was

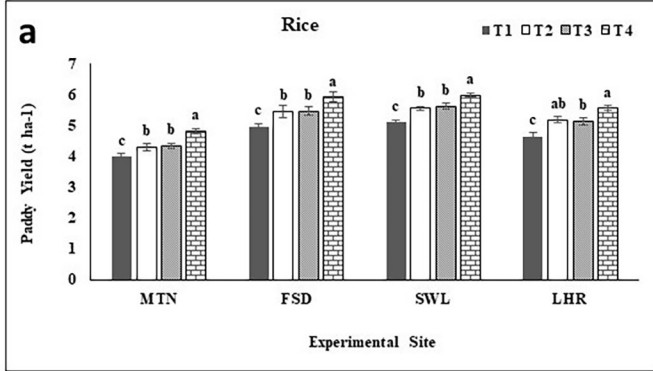 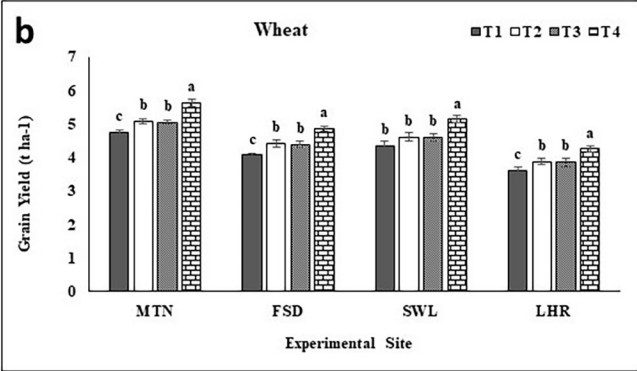

**Fig 1. Effect of zinc sulfate and bioactive zinc coated urea on paddy yield of rice and wheat grain yield.** MTN = Multan; FSD = Faisalabad; SWL = Sahiwal; LHR = Lahore; BAZU = Bioactive Zn coated urea (42% N; 1% Zn); $T_1$ (Urea 46% N @ 185 kg $ha^{-1}$+ zero Zn), $T_2$ (Urea 46% N @ 185 kg $ha^{-1}$ + $ZnSO_4$ 33% Zn @ 15 kg $ha^{-1}$), $T_3$ (BAZU 42% N @ 100 kg $ha^{-1}$ + Urea 46% N @ 62 kg $ha^{-1}$ + 1% bioactive Zn @ 1.00 kg $ha^{-1}$), $T_4$ (BAZU 42% N @ 125 kg $ha^{-1}$ + Urea 46% N @ 62 kg $ha^{-1}$ + 1% bioactive Zn @ 1.25 kg $ha^{-1}$).

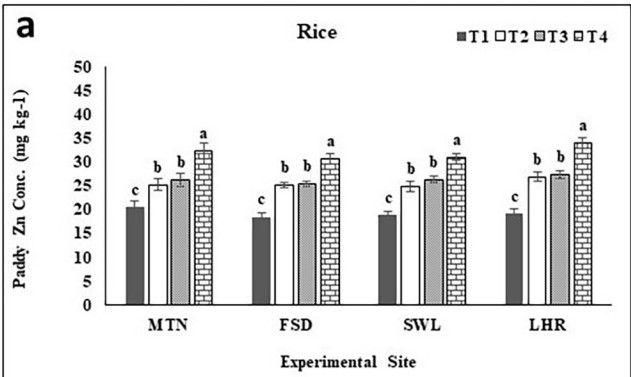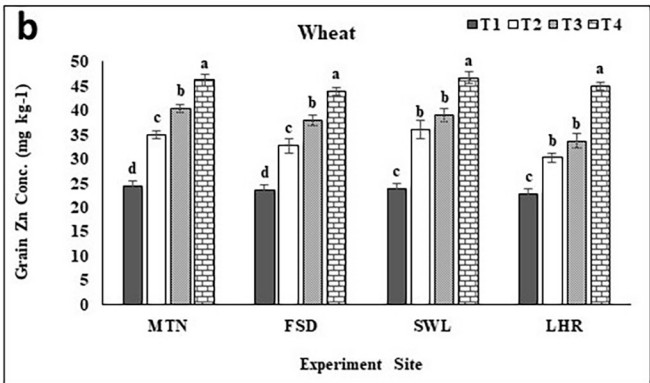

**Fig 2. Effect of zinc sulfate and bioactive zinc coated urea on Zn concentration in paddy of rice and grains of wheat.** MTN = Multan; FSD = Faisalabad; SWL = Sahiwal; LHR = Lahore; BAZU = Bioactive Zn coated urea (42% N; 1% Zn); $T_1$ (Urea 46% N @ 185 kg ha$^{-1}$+ zero Zn), $T_2$ (Urea 46% N @ 185 kg ha$^{-1}$ + ZnSO$_4$ 33% Zn @ 15 kg ha$^{-1}$), $T_3$ (BAZU 42% N @ 100 kg ha$^{-1}$ + Urea 46% N @ 62 kg ha$^{-1}$ + 1% bioactive Zn @ 1.00 kg ha$^{-1}$), $T_4$ (BAZU 42% N @ 125 kg ha$^{-1}$ + Urea 46% N @ 62 kg ha$^{-1}$ + 1% bioactive Zn @ 1.25 kg ha$^{-1}$).

recovered by plants grown in BAZU ($T_4$) applied plots (9.4, 6.6, 7.4, and 7.1-folds higher) over $T_2$ (0.51, 0.83, 0.79, and 0.89%) at MTN, FSD, SWL, and LHR respectively. A similar trend was observed in the case of agronomic efficiency i.e. relative to $T_2$ (48, 56, 66, and 52 kg kg$^{-1}$), 130, 98, 84, and 113% increment was noted under $T_4$ at MTN, FSD, SWL and LHR respectively (Fig 4C).

## Wheat

**Growth and yield attributes.** Table 3 represents the results of spike length (SL), number of grains per spike (GS), number of tillers, 1000 grain weight, biomass yield, and harvest index (HI). As compared to $T_1$ of the respective locations, SL was recorded higher at MTN (16%) and SWL (13%) under $T_4$ while at FSD and LHR, SL remained statistically unchanged among treatments. Relative to control ($T_1$), GS were significantly higher in $T_4$ plants while unchanged under $T_2$ and $T_3$ compared to $T_1$ of the respective location. Relative to $T_1$ of each location, GS were enhanced by 19, 10, 13, and 12% in presence of BAZU ($T_4$) at MTN, FSD, SWL, and LHR

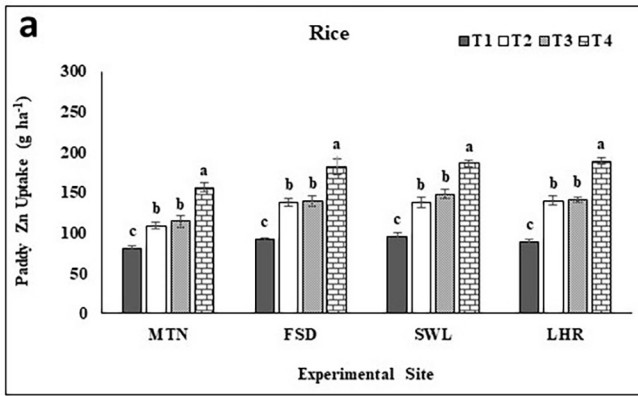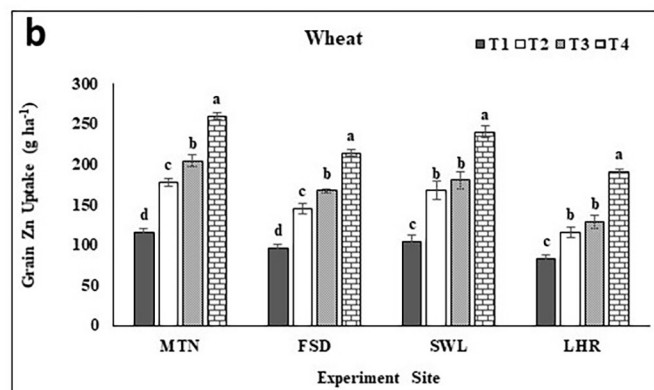

**Fig 3. Effect of zinc sulfate and bioactive zinc coated urea on Zn uptake by paddy of rice and grains of wheat.** MTN = Multan; FSD = Faisalabad; SWL = Sahiwal; LHR = Lahore; BAZU = Bioactive Zn coated urea (42% N; 1% Zn); $T_1$ (Urea 46% N @ 185 kg ha$^{-1}$+ zero Zn), $T_2$ (Urea 46% N @ 185 kg ha$^{-1}$ + ZnSO$_4$ 33% Zn @ 15 kg ha$^{-1}$), $T_3$ (BAZU 42% N @ 100 kg ha$^{-1}$ + Urea 46% N @ 62 kg ha$^{-1}$ + 1% bioactive Zn @ 1.00 kg ha$^{-1}$), $T_4$ (BAZU 42% N @ 125 kg ha$^{-1}$ + Urea 46% N @ 62 kg ha$^{-1}$ + 1% bioactive Zn @ 1.25 kg ha$^{-1}$).

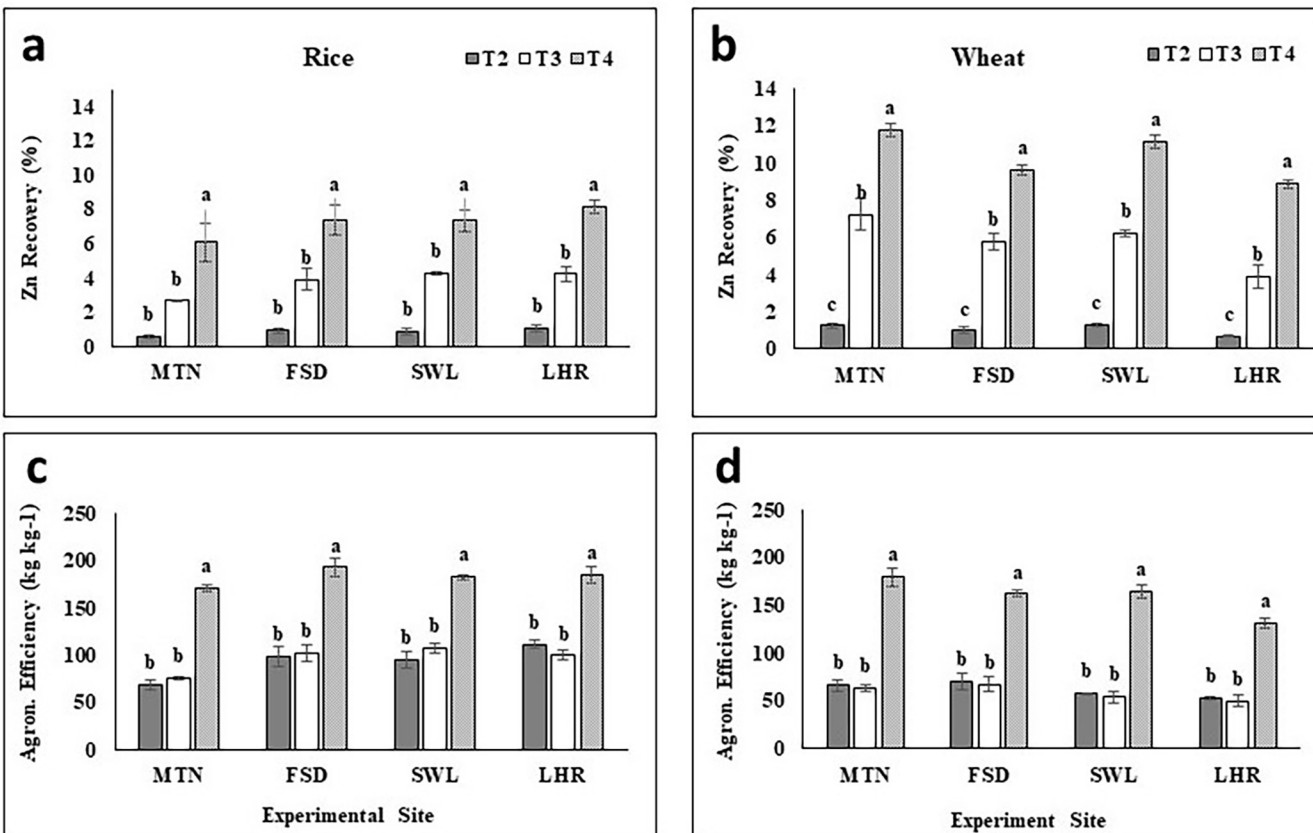

**Fig 4. Effect of zinc sulfate and bioactive zinc coated urea on agronomic and Zn recovery efficiencies of rice and wheat.** MTN = Multan; FSD = Faisalabad; SWL = Sahiwal; LHR = Lahore; BAZU = Bioactive Zn coated urea (42% N; 1% Zn); $T_1$ (Urea 46% N @ 185 kg ha$^{-1}$+ zero Zn), $T_2$ (Urea 46% N @ 185 kg ha$^{-1}$ + ZnSO$_4$ 33% Zn @ 15 kg ha$^{-1}$), $T_3$ (BAZU 42% N @ 100 kg ha$^{-1}$ + Urea 46% N @ 62 kg ha$^{-1}$ + 1% bioactive Zn @ 1.00 kg ha$^{-1}$), $T_4$ (BAZU 42% N @ 125 kg ha$^{-1}$ + Urea 46% N @ 62 kg ha$^{-1}$ + 1% bioactive Zn @ 1.25 kg ha$^{-1}$).

respectively. The BAZU ($T_4$) resulted in higher productive tillers i.e. 14, 13, 11, and 14% higher tillers were recorded at MTN, FSD, SWL, and LHR respectively over control ($T_1$) of the respective location. 1000-grain weight was significantly increased in the presence of Zn over control of each location and recorded highest under $T_4$ over the rest of treatments i.e., 8, 7, 9, and 7% higher at MTN, FSD, SWL, and LHR respectively over control ($T_1$) of the respective location. 1000-grain weight was noted as similar among $T_2$ and $T_3$ at all locations but higher than the control of each location. Similarly, grain yield was significantly increased by the application of Zn sources over control ($T_1$) of each location i.e., 12, 11, 11, and 10% higher yield was noted at MTN, FSD, SWL, and LHR respectively compared to $T_1$ of respective location ([Fig 1B]). The highest grain yield was observed under $T_4$ (5.36, 4.57, 4.90, and 4.07 t ha$^{-1}$) and lowest under $T_1$ (4.80, 4.10, 4.41, and 3.70 t ha$^{-1}$) at MTN, FSD, SWL, and LHR respectively. Grain yield was similar among $T_2$ and T3 relative to $T_1$ of each experimental location. Among locations, the highest grain yield was recorded at MTN followed by SWL, FSD, and LHR ([Fig 1B]). Harvest index (HI) followed a similar trend i.e., $T_4$ plants produced 3, 4, 4, and 4% higher HI at MTN, FSD, SWL, and LHR respectively as compared to $T_1$ of the respective location while $T_2$ and $T_3$ showed an equal increase in HI over $T_1$ ([Table 3]).

**Grain Zn concentration and uptake.** Grain Zn concentration and uptake were significantly increased by the application of Zn and the maximum concentration of Zn in grains and its uptake were noted under BAZU ($T_4$) and both were minimum in control ($T_1$) without Zn

**Table 3. Effect of zinc sulfate and bioactive zinc coated urea on growth and yield of wheat.**

| Region | Treatment | Spike Length (cm) | No. Of Grains per spike | Productive tiller per $m^2$ | 1000 grain weight (g) | Biomass yield (t $ha^{-1}$) | Harvest Index (%) |
|---|---|---|---|---|---|---|---|
| Multan | T1 | *10.18±0.17[efg] | 43.0±1.00[d] | 377±10.3[defg] | 29.10±0.25[ij] | 10.95±0.10[c] | 43.8±0.06[d] |
| | T2 | 10.6±0.16[de] | 47.6±1.20[c] | 423±12.1[abc] | 30.77±0.24[gh] | 11.35±0.13[b] | 44.7±0.12[c] |
| | T3 | 11.0±0.46[cd] | 47.3±0.88[c] | 418±9.39[abc] | 30.63±0.14[h] | 11.32±0.11[b] | 44.7±0.14[c] |
| | T4 | 11.8±0.26[ab] | 51.0±1.00[bc] | 432±13.7[ab] | 31.47±0.12[g] | 11.83±0.19[a] | 45.3±0.20[b] |
| Faisalabad | T1 | 9.80±0.17[fgh] | 39.0±0.57[e] | 337±8.41[h] | 37.10±0.21[c] | 9.50±0.13[i] | 43.3±0.07[e] |
| | T2 | 10.0±0.12[fg] | 41.0±1.00[de] | 367±12.4[fgh] | 38.50±0.44[b] | 9.88±0.20[h] | 43.5±0.19[de] |
| | T3 | 9.90±0.17[fg] | 41.5±1.44[de] | 373±11.0[efg] | 38.30±0.44[b] | 9.79±0.19[h] | 43.8±0.11[d] |
| | T4 | 10.2±0.23[ef] | 43.0±1.15[d] | 382±8.08[defg] | 39.80±0.23[a] | 10.14±0.15[fg] | 45.1±0.09[bc] |
| Sahiwal | T1 | 10.7±0.14[cde] | 48.0±1.53[c] | 399±11.5[cdef] | 27.23±0.20[k] | 10.05±0.17[g] | 43.9±0.06[d] |
| | T2 | 11.0±0.23[cd] | 51.0±1.55[bc] | 408±17.3[bcd] | 28.43±0.20[j] | 10.31±0.19[e] | 44.8±0.12[c] |
| | T3 | 11.3±0.17[bc] | 50.6±2.33[bc] | 417±14.4[abc] | 28.40±0.15[j] | 10.25±0.22[ef] | 44.9±0.04[bc] |
| | T4 | 12.1±0.13[a] | 54.0±1.15[ab] | 443±11.5[a] | 29.67±0.27[i] | 10.72±0.19[d] | 45.7±0.15[a] |
| Lahore | T1 | 9.29±0.11[h] | 51.0±1.53[bc] | 353±13.3[gh] | 33.40±0.20[f] | 8.88±0.20[k] | 41.7±0.15[g] |
| | T2 | 9.62±0.09[fgh] | 52.0±1.73[b] | 373±10.1[efg] | 34.60±0.17[e] | 9.13±0.16[j] | 42.5±0.07[f] |
| | T3 | 9.60±0.21[gh] | 52.0±1.45[b] | 384±10.5[defg] | 34.50±0.21[e] | 9.07±0.17[j] | 42.6±0.08[f] |
| | T4 | 9.8±0.17[fgh] | 57.0±1.53[a] | 403±12.3[bcde] | 35.90±0.29[d] | 9.41±0.16[i] | 43.3±0.24[e] |
| | LSD T×L | 0.595 | 3.673 | 32.53 | 0.733 | 0.163 | 0.404 |

*Average of two years data; ± Standard error; BAZU = Bioactive Zn coated urea (42% N; 1% Zn); $T_1$ (Urea 46% N @ 185 kg $ha^{-1}$+ zero Zn), $T_2$ (Urea 46% N @ 185 kg $ha^{-1}$ + $ZnSO_4$ 33% Zn @ 15 kg $ha^{-1}$), $T_3$ (BAZU 42% N @ 100 kg $ha^{-1}$ + Urea 46% N @ 62 kg $ha^{-1}$ + 1% bioactive Zn @ 1.00 kg $ha^{-1}$), $T_4$ (BAZU 42% N @ 125 kg $ha^{-1}$ + Urea 46% N @ 62 kg $ha^{-1}$ + 1% bioactive Zn @ 1.25 kg $ha^{-1}$); T = Treatment; L = Location.

fertilization (Figs 2B and 3B). Grain Zn concentration was analyzed highest in $T_4$ (46.2, 43.9, 46.7, and 44.9 mg $kg^{-1}$) and lowest in $T_1$ plants (24.3, 23.5, 23.8, and 22.8 mg $kg^{-1}$) at MTN, FSD, SWL, and LHR, respectively (Fig 2B). Grain Zn concentration was 90, 87, 96, and 97% higher at MTN, FSD, SWL, and LHR relative to $T_1$ of the respective location. At SWL, $T_2$ and $T_3$ showed an equal increase in grain Zn concentration over $T_1$. A similar trend was observed for grain Zn uptake i.e. $T_4$ plants gave maximum (248, 201, 229, and 183 g $ha^{-1}$) while minimum uptake was noted under $T_1$ (117, 97, 105, and 84 g $ha^{-1}$) at MTN, FSD, SWL, and LHR, respectively (Fig 3B). Zn uptake was increased by 112, 108, 118, and 117% over $T_1$ of the respective location whereas similar among $T_2$ and $T_3$ at SWL and LHR compared to $T_1$ (Fig 3B).

**Apparent Zn recovery and agronomic efficiency.** A significant increase in recovery of Zn in wheat grains was noted by the application of BAZU ($T_4$) as compared to $T_2$ and $T_3$ (Fig 4B). Based on an average of two-year experiments at each experimental location, the highest Zn recovery (10.7, 8.5, 10.1, and 8.1%) was observed under $T_4$ whereas recovery was calculated as lowest under $T_2$ (1.24, 0.91, 1.27 and 0.68%) at MTN, FSD, SWL, and LHR respectively. Recovery of grain Zn followed the trend T4>T3>T2 separately for each location. Similarly, agronomic efficiency was maximum under $T_4$ over the rest of the treatments while similar among $T_2$ and $T_3$ (Fig 4D). Agronomic efficiency was increased by 102, 141, 133, and 109% under $T_4$ at MTN, FSD, SWL, and LHR respectively as compared to $T_2$ of respective experimental locations (Fig 4D).

**Heatmap.** Comparison among experimental sites and treatment for both crops is presented in the form of a heat map (Figs 5 and 6). The positive correlation for all parameters was observed for $T_4$ at LHR and SWL. The $T_2$ and $T_3$ at LHR showed a positive correlation for parameters except for 1000 grain weight and harvest index whereas, at the SWL location, $T_2$

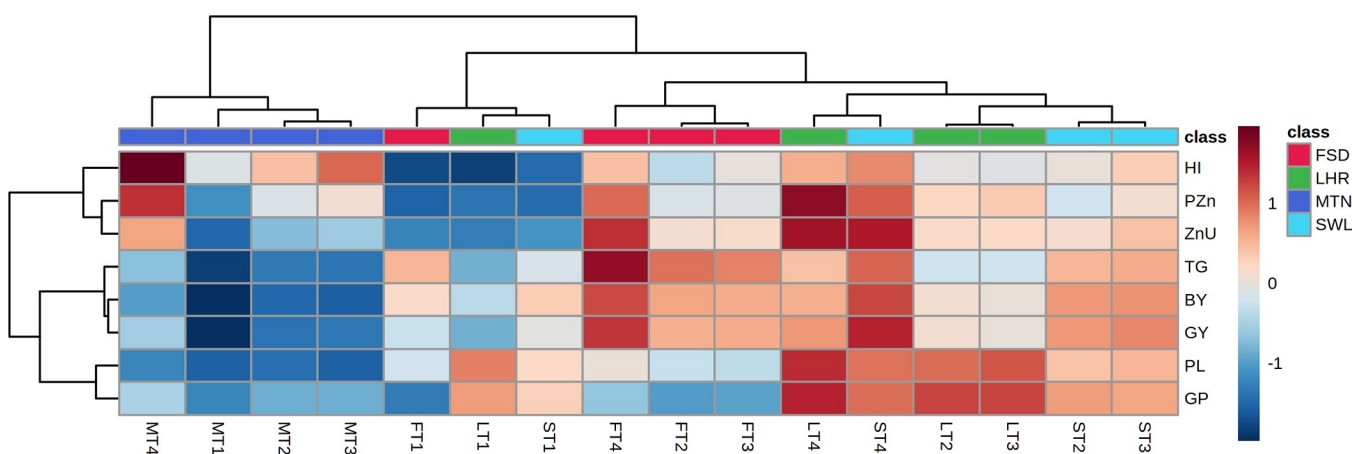

**Fig 5. Heatmap showing comparison of treatments among various experimental locations for rice.** M = Multan; F = Faisalabad; S = Sahiwal; L = Lahore; BAZU = Bioactive Zn coated urea (42% N; 1% Zn); $T_1$ (Urea 46% N @ 185 kg ha$^{-1}$ + zero Zn), $T_2$ (Urea 46% N @ 185 kg ha$^{-1}$ + ZnSO$_4$ 33% Zn @ 15 kg ha$^{-1}$), $T_3$ (BAZU 42% N @ 100 kg ha$^{-1}$ + Urea 46% N @ 62 kg ha$^{-1}$ + 1% bioactive Zn @ 1.00 kg ha$^{-1}$), $T_4$ (BAZU 42% N @ 125 kg ha$^{-1}$ + Urea 46% N @ 62 kg ha$^{-1}$ + 1% bioactive Zn @ 1.25 kg ha$^{-1}$)HI = Harvest index; PZn = paddy Zn concentration; ZnU = Zn uptake by paddy; TG = thousand paddy weight; BY = biomass yield; GY = paddy yield; PL = panicle length; GP = number of grains per panicle.

showed a negative correlation for paddy Zn concentration and $T_3$ for harvest index (Fig 5). Except for grain per panicle, $T_4$ at FSD showed a positive correlation for recorded parameters while $T_4$ at MTN showed a positive correlation for harvest index, paddy Zn concentration, and uptake (Fig 5). In wheat, a maximum positive correlation was noticed in the presence of $T_4$ at MTN followed by $T_4$ at SWL for all parameters except 1000 grain weight which remains negatively correlated for both locations (Fig 6). Except for 1000 grain weight, a positive correlation was noted for $T_2$ and $T_3$ at SWL and MTN. The $T_2$ and $T_3$ at MTN showed a higher positive correlation for grain and biological yield. A negative correlation was observed for $T_1$ at all experimental sites (Fig 6).

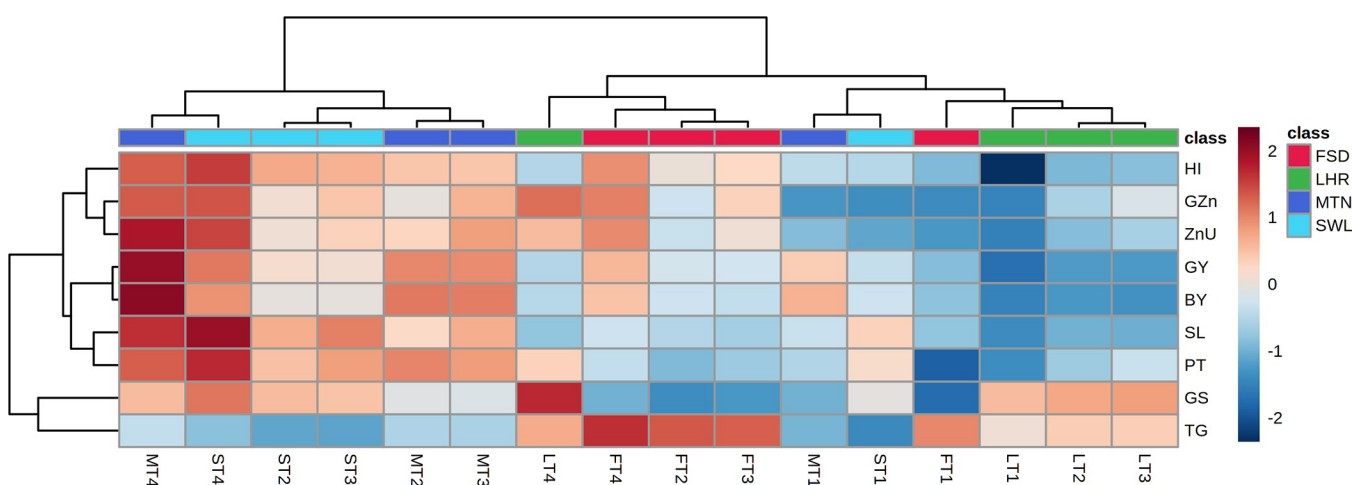

**Fig 6. Heatmap showing comparison of treatments among various experimental locations for wheat.** M = Multan; F = Faisalabad; S = Sahiwal; L = Lahore; BAZU = Bioactive Zn coated urea (42% N; 1% Zn); $T_1$ (Urea 46% N @ 185 kg ha$^{-1}$ + zero Zn), $T_2$ (Urea 46% N @ 185 kg ha$^{-1}$ + ZnSO$_4$ 33% Zn @ 15 kg ha$^{-1}$), $T_3$ (BAZU 42% N @ 100 kg ha$^{-1}$ + Urea 46% N @ 62 kg ha$^{-1}$ + 1% bioactive Zn @ 1.00 kg ha$^{-1}$), $T_4$ (BAZU 42% N @ 125 kg ha$^{-1}$ + Urea 46% N @ 62 kg ha$^{-1}$ + 1% bioactive Zn @ 1.25 kg ha$^{-1}$)GZn = Grain Zn concentration; ZnU = Zn uptake by grain; GY = grain yield; BY = biomass yield; PT = productive tillers; SL = spike length; HI = Harvest index; GS = number of grains per spike; TG = thousand grain weight.

**Table 4. Effect of zinc sulfate and bioactive zinc coated urea on economics of rice and wheat crops at various locations.**

| Region | Treatment | Grain Yield (t ha$^{-1}$) | Straw Yield (t ha$^{-1}$) | Grain Value ($ ha$^{-1}$) | Straw Value ($ ha$^{-1}$) | Gross Income ($ ha$^{-1}$) | Variable Cost ($ ha$^{-1}$) | Total Cost ($ ha$^{-1}$) | Net Benefit ($ ha$^{-1}$) | Benefit: Cost Ratio |
|---|---|---|---|---|---|---|---|---|---|---|
| | | | | Rice | | | | | | |
| Multan | T1 | *3.98[i] | 5.61[f] | 1356[i] | 64[f] | 1420[i] | 53 | 791 | 629[i] | 0.79[k] |
| | T2 | 4.21[h] | 5.71[f] | 1435[h] | 65[f] | 1500[h] | 70 | 808 | 692[h] | 0.85[jk] |
| | T3 | 4.18[h] | 5.68[f] | 1425[h] | 65[f] | 1490[h] | 56 | 794 | 696[h] | 0.87[j] |
| | T4 | 4.51[g] | 5.92[f] | 1537[g] | 67[f] | 1604[g] | 60 | 798 | 806[g] | 1.01[i] |
| Faisalabad | T1 | 4.98[e] | 7.62[bcd] | 1698[e] | 87[bcd] | 1785[e] | 53 | 791 | 994[e] | 1.25[fg] |
| | T2 | 5.26[cd] | 7.79[abc] | 1793[cd] | 89[abc] | 1882[cd] | 70 | 808 | 1074[cd] | 1.33[def] |
| | T3 | 5.31[c] | 7.67[bc] | 1810[c] | 87[bc] | 1897[c] | 56 | 794 | 1103[c] | 1.39[cd] |
| | T4 | 5.52[b] | 7.93[ab] | 1882[b] | 90[ab] | 1972[ab] | 60 | 798 | 1174[ab] | 1.47[ab] |
| Sahiwal | T1 | 5.10[de] | 7.70[abc] | 1739[de] | 87[abc] | 1826[de] | 53 | 791 | 1035[de] | 1.31[ef] |
| | T2 | 5.42[bc] | 7.79[abc] | 1848[bc] | 88[abc] | 1936[bc] | 70 | 808 | 1128[bc] | 1.39[cd] |
| | T3 | 5.36[bc] | 7.76[abc] | 1827[bc] | 88[abc] | 1915[bc] | 56 | 794 | 1121[bc] | 1.41[bc] |
| | T4 | 5.69[a] | 8.01[a] | 1940[a] | 91a | 2031[a] | 60 | 798 | 1233[a] | 1.54[a] |
| Lahore | T1 | 4.75[f] | 7.15[e] | 1619[f] | 81[e] | 1700[f] | 53 | 791 | 909[f] | 1.15[h] |
| | T2 | 5.00[e] | 7.29[de] | 1704[e] | 83[de] | 1787[e] | 70 | 808 | 979[e] | 1.21[gh] |
| | T3 | 4.95[e] | 7.23[e] | 1687[e] | 82[e] | 1769[e] | 56 | 794 | 975[e] | 1.23[g] |
| | T4 | 5.29[c] | 7.46[cde] | 1803[c] | 85[cde] | 1888[c] | 60 | 798 | 1090[cd] | 1.36[cde] |
| | | | | Wheat | | | | | | |
| Multan | T1 | 4.80[c] | 6.15[b] | 1363[c] | 350[b] | 1713[c] | 53 | 536 | 1177[c] | 2.19[b] |
| | T2 | 5.08[b] | 6.28[ab] | 1442[b] | 357[ab] | 1799[b] | 70 | 553 | 1246[b] | 2.25[bc] |
| | T3 | 5.06[b] | 6.26[ab] | 1438[b] | 356[ab] | 1794[b] | 56 | 539 | 1254[b] | 2.33[b] |
| | T4 | 5.36[a] | 6.47[a] | 1524[a] | 368[a] | 1892[a] | 60 | 543 | 1349[a] | 2.48[a] |
| Faisalabad | T1 | 4.10[g] | 5.40[efgh] | 1166[g] | 307[efgh] | 1473[fg] | 53 | 536 | 937[fg] | 1.75[fg] |
| | T2 | 4.30[f] | 5.58[cdef] | 1221[f] | 317[cdef] | 1538[e] | 70 | 553 | 985[ef] | 1.78[fg] |
| | T3 | 4.29[f] | 5.50[defg] | 1218[f] | 313[defg] | 1531[ef] | 56 | 539 | 992[ef] | 1.84[ef] |
| | T4 | 4.57[de] | 5.57[cdef] | 1298[de] | 316[cdef] | 1614[d] | 60 | 543 | 1071[d] | 1.97[d] |
| Sahiwal | T1 | 4.41[ef] | 5.64[cde] | 1252[ef] | 320[cde] | 1572[de] | 53 | 536 | 1036[de] | 1.93[de] |
| | T2 | 4.62[d] | 5.69[cd] | 1313[d] | 323[cd] | 1636[d] | 70 | 553 | 1083[d] | 1.96[d] |
| | T3 | 4.60[d] | 5.64[cde] | 1308[d] | 321[cde] | 1629[d] | 56 | 539 | 1090[d] | 2.02[d] |
| | T4 | 4.90[bc] | 5.82[c] | 1393[bc] | 331[c] | 1724[c] | 60 | 543 | 1181[c] | 2.17[c] |
| Lahore | T1 | 3.70[i] | 5.18[h] | 1052[i] | 294[h] | 1346[i] | 53 | 536 | 810[i] | 1.51[i] |
| | T2 | 3.88[h] | 5.25[gh] | 1102[h] | 298[gh] | 1400[hi] | 70 | 553 | 847[i] | 1.53[i] |
| | T3 | 3.86[hi] | 5.21[h] | 1098[hi] | 296[h] | 1394[i] | 56 | 539 | 855[hi] | 1.59[hi] |
| | T4 | 4.07[g] | 5.34[fgh] | 1157[g] | 303[fgh] | 1460[gh] | 60 | 543 | 917[gh] | 1.69[gh] |

*Average of two years; BAZU = Bioactive Zn coated urea; T$_1$ (Urea 46% N @ 185 kg ha$^{-1}$+ zero Zn), T$_2$ (Urea 46% N @ 185 kg ha$^{-1}$ + ZnSO$_4$ 33% Zn @ 15 kg ha$^{-1}$), T$_3$ (BAZU 42% N @ 100 kg ha$^{-1}$ + Urea 46% N @ 62 kg ha$^{-1}$ + 1% bioactive Zn @ 1.00 kg ha$^{-1}$), T$_4$ (BAZU 42% N @ 125 kg ha$^{-1}$ + Urea 46% N @ 62 kg ha$^{-1}$ + 1% bioactive Zn @ 1.25 kg ha$^{-1}$); $ = US dollar = 176 Pakistani Rupee; Rice paddy price = 2400 PKR per 40 kg; Rice straw price = 80 PKR per 40 kg; Wheat grain price = 2000 PKR per 40 kg; Wheat straw price = 400 PKR per 40 kg.

**Benefit-Cost ratio.** According to BCR, T$_4$ in both crops remained economical and provide higher benefits as compared to the remaining treatments (Table 4). The BCR for both crops showed the following trend T$_4$> T$_3$> T$_2$> T$_1$. In rice, the highest benefit was attained at FSD where T$_4$ gave a B:C ratio of 1.47 over 1.25 of T$_1$. Similarly, T$_4$ found economical during wheat where the B:C ratio was 2.48, 1.97, 2.17, and 1.69 while a lower B:C ratio was noted under T$_1$ (2.19, 1.75, 1.93, and 1.51) at MTN, FSD, SWL, and LHR respectively (Table 4).

## Discussion

Zinc (Zn) application, as $ZnSO_4$ and Zn enriched urea, increases grain yield in wheat [27–29] and rice [28, 30]. However, the enhancement of yield, by the application of Zn [57] through different Zn sources at various experimental sites proved to be differential in this study. The fertilization of Zn as BAZU ($T_4$) enhanced the paddy yield of rice and wheat grain yield along with the Zn concentration and uptake in comparison to $ZnSO_4$ ($T_2$). These paddy and wheat grain yield increments were linked with the number of grains per panicle and per spike, respectively, which ultimately enhanced the respective rice paddy and wheat grain weight (Tables 2 and 3), somehow, due to Zn supportive pollination through betterments in photo-synthesis, sugar transformation [24], pollen tube development [58], pollen viability [26], flow-ering and grain formation [25]. Moreover, BAZU ($T_4$) responsive yield increments might also have resulted from the higher Zn uptake [29] and recovery along with improved agronomic efficiencies (Figs 3 and 4). Previous studies also witnessed the bio-active Zn coated urea-based enhancement in morphological, yield, and quality parameters of rice [59, 60]. However, in contrast to previous studies, the rice paddy and wheat grain Zn concentrations were increased by Zn application through both $ZnSO_4$ ($T_2$) and BAZU ($T_3$ and $T_4$) sources, but the incre-ments noted with BAZU ($T_4$), were superior to the rest of the treatments (Fig 2). Mobilization and translocation of Zn in grain is dependent on its concentration in vegetative parts of the plant, soil N status and plant type (species or cultivars) [13, 61–63]. Furthermore, the Zn solu-bilizing microbes could have helped to ensure Zn availability for longer times and decreased Zn losses in soil [1, 45, 64] under BAZU application, the underlying mechanisms are yet to be explored though. Nonetheless, the bioactive Zn and Zn solubilizing bacteria present in BAZU are known to enhance Zn bioavailability to plants through the solubilization of insoluble soil Zn fractions in rhizosphere to ensure continuous supply [29]. Zn solubilizing bacteria are also known to enhance Zn availability at grain filling stage, thus accumulating higher Zn in paddy and wheat grains [29, 61] relative to other Zn sources. Apart from source-specific effect, the crop-specific effect was also observed, whereby Zn concentration in wheat grains ($\sim$47 mg kg$^{-1}$) was found to be more than paddy Zn concentration ($\sim$34 mg kg$^{-1}$) in rice (Fig 2), perhaps, due to the Zn-fertilization based [42, 60] increment in Zn uptake and subsequent higher trans-location from straw to the grain [29] by the application of BAZU ($T_4$). Whereas, the lower paddy Zn concentration was, possibly, due to the soil micro-environment (flooding) which could have reduced the phyto-availability of the Zn manifold [21, 37]. In the present study, Zn recovery and agronomic efficiencies were recorded higher as a result of BAZU ($T_4$) application due to the increments in paddy/grain yield and Zn uptake in contrast to the application of $ZnSO_4$ (Fig 4) as reported previously where Zn recovery and agronomic efficiencies of rice were improved at two different locations of Punjab Pakistan [31]. The higher Zn use efficiency was noted in the presence of Zn and bacterial enriched urea (BAZU) over conventional $ZnSO_4$ application (Fig 4) as previously, 12-fold higher Zn use efficiency was recorded in wheat by the application of bacterial enriched urea and Zn [29]. Similarly, uptake of Zn by paddy/grains was also increased with the Zn-fertilization and calculated as maximum with the addition of BAZU followed by $ZnSO_4$ over control (Fig 3). The increases in uptake of Zn by rice paddy and wheat grains were due to the increased demand resulting from the enhanced utilization of Zn in biomass production [65, 66].

To further strengthen the aforementioned arguments, the positive correlation of $T_4$ for all the parameters of rice at Lahore, Sahiwal, Faisalabad (except the number of grains per panicle), and Multan (at least for harvest index, paddy Zn concentration, and uptake) (Fig 5) further depicted the overwhelming response of BAZU application. Similarly, except for 1000 grain weight, $T_4$ showed a highly positive correlation for all attributes at Multan and Sahiwal (Fig 6).

**Table 5. Correlation among parameters influenced by BAZU application @125 kg ha$^{-1}$ (T4) in four different sites (MTN, FSD, SWL, LHR).**

| Rice | | | | | | | | | |
|---|---|---|---|---|---|---|---|---|---|
| | BY | GP | PY | HI | PL | PZn | TG | ZnU | |
| **GP** | 0.3860 | 1 | | | | | | | |
| **PY** | **0.9975** | 0.4029 | 1 | | | | | | |
| **HI** | -0.9495 | -0.3259 | -0.9253 | 1 | | | | | |
| **PL** | **0.7624** | **0.8697** | **0.7610** | -0.7499 | 1 | | | | |
| **PZn** | -0.3930 | 0.5569 | -0.4114 | 0.2499 | 0.2665 | 1 | | | |
| **TG** | **0.9256** | 0.0224 | **0.9090** | -0.9282 | 0.4977 | -0.5882 | 1 | | |
| **ZnU** | **0.8808** | **0.7353** | **0.8733** | -0.8800 | **0.9735** | 0.0848 | 0.6813 | 1 | |
| **Wheat** | | | | | | | | | |
| | BY | GY | HI | TG | ZnU | GS | GZn | PT | SL |
| **GY** | **0.9914** | 1 | | | | | | | |
| **HI** | **0.7343** | **0.8166** | 1 | | | | | | |
| **TG** | -0.6066 | -0.6067 | -0.4530 | 1 | | | | | |
| **ZnU** | **0.9793** | **0.9870** | **0.7985** | -0.7267 | 1 | | | | |
| **GS** | -0.1627 | -0.2175 | -0.4339 | -0.5887 | -0.0696 | 1 | | | |
| **GZn** | 0.6213 | 0.6239 | 0.4777 | -0.9996 | **0.7415** | 0.5663 | 1 | | |
| **PT** | 0.6284 | 0.6329 | 0.4930 | -0.9990 | **0.7491** | 0.5527 | **0.9998** | 1 | |
| **SL** | **0.8313** | **0.8672** | **0.8229** | -0.8729 | **0.9265** | 0.1200 | **0.8861** | **0.8937** | 1 |

PZn/GZn = paddy/grain Zn concentration; ZnU = Zn uptake by paddy/grain; GY = paddy/grain yield; BY = biomass yield; PT = productive tillers; PL/SL = panicle/spike length; HI = Harvest index; GP/GS = number of grains per panicle/spike; TG = thousand paddy/grain weight.

The previous study showed a positive correlation among recorded parameters of only wheat crop by the application of Zn and bacterial enriched urea [29]. Apart from the benefit to cost ratio reported previously only for the wheat [29] the maximum benefit to cost ratio of T$_4$ in rice (1.47 vs 1.25 in T$_1$) as well as wheat (2.48 vs 2.19 in T$_1$) at Faisalabad and Multan (Table 4), respectively, reinforced the economic effectiveness BAZU (T$_4$) application. Application of BAZU (T$_4$) might have activated or regulated soil and plant mechanisms in our study leading to impart its beneficial impacts on the growth, yield, Zn uptake, Zn recovery, and agronomic efficiencies of both crops (rice-wheat). In addition, the enhancements in rice and wheat growth and yield in this study would have, possibly, contributed by the collective responses of bioactive-Zn and supporting activities of microbes involved in P-solubilization, ACC deaminase activity, production of siderophores and indole-3-acetic acid [67–70] Finally, the strongly positive interaction (Pearson correlation) of Zn-uptake with rice (biomass yield, number of grains per panicle, paddy yield, and panicle length) and wheat (biomass yield, grain yield, harvest index, grain Zn concentration, productive tillers, and spike length) parameters highlighted the continuous Zn availability to both crops enabling them to uptake Zn and utilize it in the subsequent biomass and yield production attributes (Table 5), the underlying molecular mechanisms of which are still needed to be investigated in future studies.

## Conclusion

The application of Zn either as sole ZnSO$_4$ or as bioactive zinc-coated urea (BAZU) increased paddy and wheat grain yield and Zn concentration however, the increments were found to be the highest with 125 kg BAZU per hectare. Among test crops, the maximum grain Zn concentration was analyzed in wheat followed by rice. Grain yield was recorded in the order BAZU @ 125 kg ha$^{-1}$ > ZnSO$_4$ = BAZU @ 100 kg ha$^{-1}$ > Control for both crops. In this study, the

superiority of BAZU @ 125 kg ha$^{-1}$ over ZnSO$_4$ in terms of grain Zn concentration can make it an effective choice for the biofortification of Zn in the rice paddy and wheat grain. However, the underlying physiological and molecular mechanisms can be further investigated in the future.

## Acknowledgments

The authors highly acknowledge Engro Fertilizers Ltd. for the provision of experimental funds. We also acknowledge Field assistants (FAs) Muhammad Naseer Iqbal, Muhammad Shahid, Rao Ali Hasan (FAs Lahore), Mohsin Shabir, Kashif Murtaza (FAs Multan), Muhammad Tayyab, Muhammad Mubeen Javed (FAs Sahiwal), Muhammad Abid, Ahmad Nawaz (FAs Faisalabad) and Kanwaer Abdul Khalique, Muhammad Faseeh (FAs Hyderabad) of Engro fertilizers R&D to help in the trials execution and data collection, Laboratory incharge Muhammad Tariq and Rashid Mahmood, lab assistant Awais Afzal, Maria Mahmood, Arif Ahmed Sheikh and lab attendant Muhammad Kashif and Shahzad Mughal for soil and plant samples preparation and analysis. The authors are grateful to Rice Research Institute Kala Shah Kaku and Wheat Research Institute Ayub Agricultural Research Institute Faisalabad for the provision of rice and wheat seeds respectively. We acknowledge Bilal Aziz R&D coordinator Engro Fertilizers for the formatting of the draft.

## Author Contributions

**Conceptualization:** Syed Shahid Hussain Shah, Muhammad Asif Ali.

**Data curation:** Muhammad Azhar, Muhammad Yasir Khurshid, Muhammad Hasnain, Zeeshan Ali.

**Formal analysis:** Ijaz Ahmad.

**Investigation:** Muhammad Yasir Khurshid, Muhammad Hasnain, Zeeshan Ali, Ahmad Abu Al-Ala Shaheen.

**Methodology:** Muhammad Azhar, Muhammad Yasir Khurshid.

**Project administration:** Muhammad Asif Ali.

**Resources:** Muhammad Asif Ali.

**Software:** Faisal Nadeem.

**Supervision:** Syed Shahid Hussain Shah.

**Validation:** Muhammad Azhar, Muhammad Naeem Khan, Muhammad Hasnain, Zeeshan Ali.

**Visualization:** Muhammad Azhar, Muhammad Yasir Khurshid, Muhammad Hasnain, Zeeshan Ali, Ahmad Abu Al-Ala Shaheen.

**Writing – original draft:** Muhammad Azhar.

**Writing – review & editing:** Faisal Nadeem, Muhammad Naeem Khan.

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
