## [Decision Letter · Decision Letter 0]

15 Nov 2022

PONE-D-22-26698Enhancements in yield, agronomic and zinc recovery efficiencies of rice-wheat system through bioactive zinc coated urea application in AridisolsPLOS ONE

Dear Dr. Azhar,

Thank you for submitting your manuscript to PLOS ONE. After careful consideration, we feel that it has merit but does not fully meet PLOS ONE’s publication criteria as it currently stands. Therefore, we invite you to submit a revised version of the manuscript that addresses the points raised during the review process.

We look forward to receiving your revised manuscript.

Kind regards,

Min Huang

Academic Editor

PLOS ONE

Journal Requirements:

"No Funding was received for study"

We note that one or more of the authors are employed by a commercial company: Engro Fertilizers Limited

Reviewers' comments:

Reviewer's Responses to Questions

**Comments to the Author**

1. Is the manuscript technically sound, and do the data support the conclusions?

Reviewer #1: Yes

Reviewer #2: Yes

2. Has the statistical analysis been performed appropriately and rigorously? 

Reviewer #1: Yes

Reviewer #2: Yes

3. Have the authors made all data underlying the findings in their manuscript fully available?

Reviewer #1: No

Reviewer #2: Yes

4. Is the manuscript presented in an intelligible fashion and written in standard English?

Reviewer #1: Yes

Reviewer #2: Yes

5. Review Comments to the Author

Reviewer #1: This is a very interesting paper to investigate the Enhancements in yield, agronomic and zinc recovery efficiencies of rice-wheat system through bioactive zinc coated urea application in Aridisols. The effects of bioactive zinc coated urea are promising. However, the discussion is not enough. In addition, average of two years results were shown, however the results of each year were not shown. The interactions of year, sites and treatments were not clear. The Pearson correlation relationships among different agronomic parameters were not investigated. The novelty, in-depth of data analysis, and global relevance of the paper should be strengthen. Therefore, I suggest a moderate revision.

Reviewer #2: I have gone through the manuscript “Enhancements in yield, agronomic and zinc recovery efficiencies of rice-wheat system through bioactive zinc coated urea application in Aridisols”. I have following observations:

Overall, a nicely structured abstract. The most important data to add is the grain Zn concentration data as this is the key data when dealing with efficiencies of Zn. only comparative percentage values are provided here. It will be good to present quantitative data to present absolute impact of treatments applied.

20-21: adopt a similar treatment caption throughout MS and add Kg ha-1 at each dose.

29-30: Zinc recovery upto 9-11 folds is too high. Explain the causes of why Zinc such a high value especially when Zn source was also applied in Treatment T2 through Zn sulphate. Please revise

44: There are lot of research have been done since 2003. Please review the literature, add updated data and source.

46: Please avoid citing old references wherever possible. Cite new work on similar lines.

More information on soil type, texture, soil depth and nutrient status of soils, at least of the plough layer may be added.

118-119 Field remained flooded (∼10 cm depth) for one week after seedling transplantation, drained after one week and refilled (∼10 cm depth).....why?

Please, provide the information on how you did test the requirements of an ANOVA (homogeneity of variances and normal distribution of the residuals). Also provide information how you treated the data if the requirements were not fulfilled (data transformation?).

What about the system productivity? How the yields and economics of rice-wheat in a system mode was influenced by different treatments.

Authors have got funding from a private organization for evaluating their product and they are reporting the product as best too. This is clearly a matter of conflict of interests/ competing interests. But the authors declare that they have no competing interests. As far as I understand, they should declare it as a matter of competing interests.

Methodology and Results chapters are written well, but discussion part needs improvement. Please revise.

Table 1; ppm is not the SI unit. Please report it as mg/kg.

Table 3: Productive tillers? unit is missing. Is it in m2?

Fig. 1: In the fig. heading you are writing paddy yield, whereas in the figure it is rice yield? Please correct it, keep uniformity throughout the manuscript.

Fig. 2 & 3: Same as fig. 1. Even within figure there are 2 terms. How did you carry out the analysis for Zn concentration and uptake? Rice grain with or without husk?

Overall, the manuscript is written well, but novelty is explicitly not mentioned in the introduction chapter. Considering the importance of the subject and information generated in realistic on-farm scenario, the authors may be given a chance to improve the manuscript.

6. PLOS authors have the option to publish the peer review history of their article (what does this mean?). If published, this will include your full peer review and any attached files.

Reviewer #1: No

Reviewer #2: **Yes: **RS Bana

---

## [Author Response · Author response to Decision Letter 0]

26 Dec 2022

Editors Comments

Response: A separate file labelled 'Response to Reviewers' was uploaded

Response: A separate file labelled 'Revised Manuscript with Track Changes' was uploaded

Response: An unmarked version of revised paper (without tracked changes) was uploaded as a separate file labelled 'Manuscript'.

Journal Requirements

Response: Manuscript is in line with PLOS ONE’s style requirements.

Response: Authors roles/contribution have been updated in the submission form.

“The funder provided support in the form of salaries for authors [insert relevant initials] but did not have any additional role in the study design, data collection and analysis, decision to publish, or preparation of the manuscript. The specific roles of these authors are articulated in the ‘author contributions’ section.” If your commercial affiliation did play a role in your study, please state, and explain this role within your updated Funding Statement.

Response: Funding statement has been updated with required information.

4. Please also provide an updated Competing Interests Statement declaring this commercial affiliation along with any other relevant declarations relating to employment, consultancy, patents, products in development, or marketed products, etc. Within your Competing Interests Statement, please confirm that this commercial affiliation does not alter your adherence to all PLOS ONE policies on sharing data and materials by including the following statement: "This does not alter our adherence to PLOS ONE policies on sharing data and materials.”(as detailed online in our guide for authors http://journals.plos.org/plosone/s/competing-interests). If this adherence statement is not accurate and there are restrictions on sharing of data and/or materials, please state these. Please note that we cannot proceed with consideration of your article until this information has been declared.

Response: Competing interest statement has been included in cover letter.

Response: As suggested, funding and competing interest statements have been added in cover letter.

6. In your Data Availability statement, you have not specified where the minimal data set underlying the results described in your manuscript can be found. PLOS defines a study's minimal data set as the underlying data used to reach the conclusions drawn in the manuscript and any additional data required to replicate the reported study findings in their entirety. All PLOS journals require that the minimal data set be made fully available. For more information about our data policy, please see http://journals.plos.org/plosone/s/data-availability. Upon re-submitting your revised manuscript, please upload your study’s minimal underlying data set as either Supporting Information files or to a stable, public repository and include the relevant URLs, DOIs, or accession numbers within your revised cover letter. For a list of acceptable repositories, please see http://journals.plos.org/plosone/s/data-availability#loc-recommended-repositories. Any potentially identifying patient information must be fully anonymized. Important: If there are ethical or legal restrictions to sharing your data publicly, please explain these restrictions in detail. Please see our guidelines for more information on what we consider unacceptable restrictions to publicly sharing data: http://journals.plos.org/plosone/s/data-availability#loc-unacceptable-data-access-restrictions. Note that it is not acceptable for the authors to be the sole named individuals responsible for ensuring data access. We will update your Data Availability statement to reflect the information you provide in your cover letter.

Response: Data availability statement has been added in cover letter.

Review comments to Author

Reviewer 1

1. This is a very interesting paper to investigate the Enhancements in yield, agronomic and zinc recovery efficiencies of rice-wheat system through bioactive zinc coated urea application in Aridisols. The effects of bioactive zinc coated urea are promising. However, the discussion is not enough. In addition, average of two-year results were shown, however the results of each year were not shown. The interactions of year, sites and treatments were not clear. The Pearson correlation relationships among different agronomic parameters were not investigated. The novelty, in-depth of data analysis, and global relevance of the paper should be strengthened. Therefore, I suggest a moderate revision.

Response: As the experiments involved four experimental sites, four treatments and two crops so, the presentation of data of two years would have not been convenient form the point of view of results description and relevant discussion. Given the fact that the trend of results remained consistent in two experimental years hence, we preferred the presentation of average data in results. As far as interaction of years, sites and treatments is concerned, we presented heat maps as Figure 5 (for rice) and Figure 6 (for wheat) to represent site-wise interactive significance treatments with various parameters studied. Pearson correlation (table 5) has been added. Discussion section has been revised with additional information (Line 287-370).

Reviewer 2

1. I have gone through the manuscript “Enhancements in yield, agronomic and zinc recovery efficiencies of rice-wheat system through bioactive zinc coated urea application in Aridisols”. I have following observations: Overall, a nicely structured abstract. The most important data to add is the grain Zn concentration data as this is the key data when dealing with efficiencies of Zn. only comparative percentage values are provided here. It will be good to present quantitative data to present absolute impact of treatments applied.

Response: As suggested, quantitative data to present Zn concentration have been added (Line 39-41). 

2. 20-21: adopt a similar treatment caption throughout MS and add Kg ha-1 at each dose. 

Response: Revised as suggested (Line 29-31)

3. 29-30: Zinc recovery upto 9-11 folds is too high. Explain the causes of why Zinc such a high value especially when Zn source was also applied in Treatment T2 through Zn sulphate. Please revise

Response: We understand the concern of the reviewer. However, we would like to explain our understanding about the Zn-recovery efficiency and its calculations. Zn recovery efficiency is calculated by subtracting the Zn uptake in control (T1) from the Zn uptake in treatment (T2, T3 or T4) divided by the quantity of Zn obtained from the relevant applied source. As ZnSO4 (T2) applied at the rate of 14.8 kg/ha gives 4890 g Zn/ha, BAZU (T3) applied at the rate of 100 kg/ha gives 1000 g Zn/ha and BAZU (T4) applied at the rate of 125 kg/ha gives 1250 g Zn/ha. Hence, the lesser values of denominators for the calculation of Zn recovery efficiency under T3 and T4, with respect to T2, gave larger differences which contributed to higher fold changes (9-11). The equation for determination of apparent Zn recovery is given in materials and method section (line number 184-185).

4. 44: There are lot of research have been done since 2003. Please review the literature, add updated data and source.

Response: As per suggestions, updated data and citations have been added (Line number 56-65)

5. 46: Please avoid citing old references wherever possible. Cite new work on similar lines.

Response: Updated References have been added (Line number 56-65)

6. More information on soil type, texture, soil depth and nutrient status of soils, at least of the plough layer may be added.

Response: As suggested, Information has been added in table 1

7. 118-119 Field remained flooded (∼10 cm depth) for one week after seedling transplantation, drained after one week and refilled (∼10 cm depth).....why?

Response: This was practiced to avoid algae growth (Farooq M, Ullah A, Rehman A, Nawaz A, Nadeem A, Wakeel A, et al. Application of zinc improves the productivity and biofortification of fine grain aromatic rice grown in dry seeded and puddled transplanted production systems. Field Crops Research. 2018; 216:53-62.) 

8. Please, provide the information on how you did test the requirements of an ANOVA (homogeneity of variances and normal distribution of the residuals). Also provide information how you treated the data if the requirements were not fulfilled (data transformation?).

Response: We understand the concern of the reviewer about the statistical analysis. We would like to explain that there were 4 treatments, 4 experimental sites, 4 replicates and two crops (wheat and rice). There were 10 and 11 parameters investigated for rice and wheat crops, respectively. The ANOVA was applied to each crop separately and their respective heatmaps were also generated separately. The statistical analysis was revised to include this information (Line number 193-194)

9. What about the system productivity? How the yields and economics of rice-wheat in a system mode was influenced by different treatments.

Response: Table 4 provides the information regarding the influence of different treatments on the economics of rice wheat production system. It has also been discussed in discussion section (Line number 349-354). 

10. Authors have got funding from a private organization for evaluating their product and they are reporting the product as best too. This is clearly a matter of conflict of interests/ competing interests. But the authors declare that they have no competing interests. As far as I understand, they should declare it as a matter of competing interests.

Response: We understand the concerns raised by the reviewer. There is no existence of any conflict of interest in this study. The research work is conducted by the research and development (R&D) wing of Engro Fertilizers Limited. Adhering strongly with the research ethics, this wing already has research publications in renowned scientific journals. In this study, we did not advocate the superiority of our product, rather, we used the term BAZU instead of our product’s trade name which removes any conflict of interest. Moreover, the abbreviation of BAZU stands for bioactive Zn-coated urea which is not a name of any of Engro’s products. This is actually the formulation and anyone in the research field whether public or private can prepare it. To further dilute the impression of reporting BAZU as best, we have revised the conclusion section as “In this study, the superiority of BAZU @ 125 kg ha-1 over ZnSO4 in terms of grain Zn concentration can make it an effective choice for the biofortification of Zn in rice paddy and wheat grain. However, the underlying physiological and molecular mechanisms can be further investigated in future (Lines 377-381). 

11. Methodology and Results chapters are written well, but discussion part needs improvement. Please revise.

Response: As suggested, Discussion part has been improved (Line number 289-372)

12. Table 1; ppm is not the SI unit. Please report it as mg/kg.

Response: Revised as suggested

13. Table 3: Productive tillers? unit is missing. Is it in m2?

Response: Yes, per m2. Unit has been added in Table 3

14. Fig. 1: In the fig. heading you are writing paddy yield, whereas in the figure it is rice yield? Please correct it, keep uniformity throughout the manuscript.

Response: Following reviewer suggestion Headings have been corrected to ensure uniformity.

15. Fig. 2 & 3: Same as fig. 1. Even within figure there are 2 terms. How did you carry out the analysis for Zn concentration and uptake? Rice grain with or without husk?

Response: The figures were revised to ensure uniformity of terms used as suggested by the reviewer. Paddy (rice grain with husk) was used for the analysis of Zn concentration and uptake.

16. Overall, the manuscript is written well, but novelty is explicitly not mentioned in the introduction chapter. Considering the importance of the subject and information generated in realistic on-farm scenario, the authors may be given a chance to improve the manuscript.

Response: Bioactive Zn coated urea (BAZU) is an emerging approach not tested for Zn recovery and agronomic efficiency. Following reviewer suggestion Novelty statement has been added in introduction section (Line 102-107)

---

## [Decision Letter · Decision Letter 1]

13 Jan 2023

PONE-D-22-26698R1Enhancements in yield, agronomic and zinc recovery efficiencies of rice-wheat system through bioactive zinc coated urea application in AridisolsPLOS ONE

Dear Dr. Azhar,

Thank you for submitting your manuscript to PLOS ONE. After careful consideration, we feel that it has merit but does not fully meet PLOS ONE’s publication criteria as it currently stands. Therefore, we invite you to submit a revised version of the manuscript that addresses the points raised during the review process.

Specifically, one reviewer raised concerns on the English language. Please note that PLOS ONE does not provide copyediting or proofs of accepted manuscripts. We therefore recommend that you carefully review your manuscript and correct any errors at this time.

We look forward to receiving your revised manuscript.

Kind regards,

Min Huang

Academic Editor

PLOS ONE

Journal Requirements:

Reviewers' comments:

Reviewer's Responses to Questions

**Comments to the Author**

1. If the authors have adequately addressed your comments raised in a previous round of review and you feel that this manuscript is now acceptable for publication, you may indicate that here to bypass the “Comments to the Author” section, enter your conflict of interest statement in the “Confidential to Editor” section, and submit your "Accept" recommendation.

Reviewer #1: All comments have been addressed

Reviewer #2: All comments have been addressed

2. Is the manuscript technically sound, and do the data support the conclusions?

Reviewer #1: Yes

Reviewer #2: Yes

3. Has the statistical analysis been performed appropriately and rigorously? 

Reviewer #1: Yes

Reviewer #2: Yes

4. Have the authors made all data underlying the findings in their manuscript fully available?

Reviewer #1: Yes

Reviewer #2: Yes

5. Is the manuscript presented in an intelligible fashion and written in standard English?

Reviewer #1: Yes

Reviewer #2: Yes

6. Review Comments to the Author

Reviewer #1: (No Response)

Reviewer #2: The authors have addressed almost all the concerns. The authors are advised to make minor typographical and English language related corrections. The manuscript may be accepted now.

7. PLOS authors have the option to publish the peer review history of their article (what does this mean?). If published, this will include your full peer review and any attached files.

Reviewer #1: No

Reviewer #2: No

---

## [Author Response · Author response to Decision Letter 1]

16 Feb 2023

Editors Comments

Response: A separate file labelled 'Response to Reviewers' was uploaded

Response: A separate file labelled 'Revised Manuscript with Track Changes' was uploaded

Response: An unmarked version of revised paper (without tracked changes) was uploaded as a separate file labelled 'Manuscript'.

Journal Requirements

Response: Reference list has been corrected and is complete now (Line # 284, 323)

Review comments to Author

Reviewer 1

No Response

Reviewer 2

The authors have addressed almost all the concerns. The authors are advised to make minor typographical and English language related corrections. The manuscript may be accepted now.

Response: As suggested, typographical and English language related corrections have been made throughout the manuscript.

---

## [Editor Report · Decision Letter 2]

20 Feb 2023

Enhancements in yield, agronomic, and zinc recovery efficiencies of rice-wheat system through bioactive zinc coated urea application in Aridisols

PONE-D-22-26698R2

Dear Dr. Azhar,

We’re pleased to inform you that your manuscript has been judged scientifically suitable for publication and will be formally accepted for publication once it meets all outstanding technical requirements.

Kind regards,

Min Huang

Academic Editor

PLOS ONE

---

## [Editor Report · Acceptance letter]

27 Feb 2023

PONE-D-22-26698R2 

Enhancements in yield, agronomic, and zinc recovery efficiencies of rice-wheat system through bioactive zinc coated urea application in Aridisols 

Dear Dr. Azhar:

I'm pleased to inform you that your manuscript has been deemed suitable for publication in PLOS ONE. Congratulations! Your manuscript is now with our production department. 

Kind regards, 

on behalf of

Dr. Min Huang 

Academic Editor

PLOS ONE